# Neutrophil Extracellular Traps in the Establishment and Progression of Renal Diseases

**DOI:** 10.3390/medicina55080431

**Published:** 2019-08-02

**Authors:** Hector Salazar-Gonzalez, Alexa Zepeda-Hernandez, Zesergio Melo, Diego Eduardo Saavedra-Mayorga, Raquel Echavarria

**Affiliations:** 1Decanato de Ciencia y Tecnología, Universidad Autónoma de Guadalajara, Zapopan 45129, Mexico; 2Facultad de Medicina, Universidad Autónoma de Guadalajara, Zapopan 45129, Mexico; 3CONACyT-Centro de Investigación Biomédica de Occidente, Instituto Mexicano del Seguro Social, Sierra Mojada #800 Col. Independencia, Guadalajara 44340, Mexico; 4Facultad de Medicina, Centro Universitario de Ciencias de la Salud, Universidad de Guadalajara, Guadalajara 44340, Mexico

**Keywords:** kidney, AKI, autoimmunity, vasculitis, neutrophils, NETosis, NETs

## Abstract

Uncontrolled inflammatory and immune responses are often involved in the development of acute and chronic forms of renal injury. Neutrophils are innate immune cells recruited early to sites of inflammation, where they produce pro-inflammatory cytokines and release mesh-like structures comprised of DNA and granular proteins known as neutrophil extracellular traps (NETs). NETs are potentially toxic, contribute to glomerular injury, activate autoimmune processes, induce vascular damage, and promote kidney fibrosis. Evidence from multiple studies suggests that an imbalance between production and clearance of NETs is detrimental for renal health. Hence strategies aimed at modulating NET-associated processes could have a therapeutic impact on a myriad of inflammatory diseases that target the kidney. Here, we summarize the role of NETs in the pathogenesis of renal diseases and their mechanisms of tissue damage.

## 1. Introduction

Renal disease and the immune system are inextricably linked. The kidneys are frequent targets of harmful immune responses that contribute to the development of acute forms of renal injury, as well as chronic kidney disease progression [1,2,3,4]. Additionally, patients experiencing kidney failure often suffer disruption of immune homeostasis; a factor that negatively impacts on their morbidity and mortality [5]. The mechanisms driving the immunopathology of renal diseases are diverse and include immune cell recruitment, autoantibody production, the formation of immune complexes, dysregulation of inflammatory mediators, and immunodeficiency [4,6]. Furthermore, activation of tissue repair mechanisms after an immune-mediated injury can lead to fibrosis and ultimately, kidney failure [7]. Basic and clinical research findings highlight the importance of innate immune cells such as neutrophils in the pathogenesis and progression of renal diseases.

Neutrophils are the first cellular effectors to be recruited at sites of infection and tissue damage; where they play a significant role in inflammation, immune cell recruitment, pathogen clearance, and tissue repair. Neutrophils execute their functions through four main mechanisms: phagocytosis, degranulation, cytokine production, and neutrophil extracellular trap (NET) formation [8,9]. A successful host defense requires efficient neutrophil activation that, in the case of sterile inflammation, is triggered by danger signals known as damage-associated molecular patterns (DAMPs). The innate immune system recognizes DAMPs after their release from damaged cells, or due to chemical and proteolytic modifications secondary to tissue injury. In the kidney, neutrophil infiltration and DAMPs released by necrotic renal cells amplify intrarenal inflammation and tissue damage [2,10]. DAMPs also enhance tubular injury by stimulating neutrophil receptors to induce activation of the enzyme peptidyl arginine deiminase 4 (PAD4), chromatin decondensation, and NET formation [11]. NETs are also a source of autoantigens that induce production of anti-neutrophil cytoplasmic antibodies (ANCAs) [12]. In this review, we summarize the ongoing knowledge regarding NET-associated mechanisms of tissue damage in the establishment and progression of kidney diseases (Figure 1).

## 2. Neutrophil Extracellular Traps

NETs are meshed extracellular structures comprised of DNA, histones, and proteins derived from polymorphonuclear granules [9]. These structures concentrate immune effectors locally and help contain pathogen dissemination. Some of the molecular effectors contributing to the microbicidal potential of NETs are neutrophil elastase (NE), myeloperoxidase (MPO), cathepsin G, lactoferrin, pentraxin 3, gelatinase, proteinase 3, and peptidoglycan binding proteins [13,14]. NE and cathepsin G can also process interleukin 1 and interleukin 36, thus emphasizing the role of NET formation in cytokine activation and inflammation [15].

Significant sequential changes preceding NET release onto the extracellular space are nuclear membrane disruption, cytoplasmic chromatin decondensation, and plasma membrane burst [9]. In this scenario, NETs emerge from dying neutrophils through a cell death mechanism that differs from apoptosis and necrosis, and is known as NETosis [16]. In suicidal NETosis cell death is a consequence of NET release. However, a different pathway known as vital NETosis can lead to NET formation while still maintaining cell viability [17]. The type of NETosis that neutrophils undertake mostly depends on the molecular nature of the stimuli [18]. A recent proteomic analysis revealed that neutrophil stimulation with phorbol 12-myristate 13-acetate (PMA), calcium ionophore A23187 or *Escherichia coli* lipopolysaccharide induce NETs with different protein compositions and post-translational modifications; features that likely reflect their biological function [19].

Suicidal NETosis relies on the intracellular increase of reactive oxygen species (ROS), which leads to NE nuclear translocation; where it partially degrades histones and triggers chromatin decondensation [16,20]. MPO synergizes with NE to produce massive chromatin relaxation [20]. Proteolysis by NE is not the only histone modification involved in NET formation. Histone hypercitrullination by PAD4 also mediates nucleosome destabilization and chromatin decondensation. Neutrophils express high levels of PAD4, an enzyme related to the hypercitrullinated histones H3 and H4 present in both, decondensed chromatin and NETs [21]. The pro-inflammatory cytokines interleukin 1 beta, tumor necrosis factor-alpha, and interleukin 8 are potent activators of ROS production in neutrophils and induce NET formation [22]. Calcium mobilization and protein kinase C (PKC) isoforms are also critical regulators of NETosis. In a coordinated balance, PKCα inhibits histone deamination; whereas PKCζ leads to PAD4 activation and histone citrullination [23]. Additionally, ROS activate mitogen-activated protein kinase p38 and downstream p38-regulated/activated kinase (PRAK) to induce NET formation in response to PMA [24]. PRAK is also an oxidative stress sensor and, along with p38, regulates the balance between NETosis and apoptosis in neutrophils.

In contrast, vital NETosis does not involve plasma membrane damage or cellular lysis since NET release occurs through budding nuclear vesicles filled with DNA [17]. Neutrophils that undergo vital NETosis become anuclear but maintain plasma membrane integrity, motility, and the ability to perform phagocytosis [25]. Partial triggers of vital NETosis are the activation of toll-like receptors (TLRs) and complement factor 3 [26]. In contrast with suicidal NETosis, this pathway is faster and mostly oxidant-independent [27]. However, a recent report described a ROS-dependent pathway that involves mitochondrial DNA and leads to vital NETosis in neutrophils previously primed with granulocyte-macrophage colony-stimulating factor and stimulated with lipopolysaccharide [28].

Autophagy, a conserved catabolic process preventing cellular damage under stress and cytotoxic insults, also regulates NET formation [29,30]. However, there are conflicting results among studies that evaluate the effect of autophagy inhibitors and activators on NET formation. Studies by *Itakura A.* et al. and *Park S.Y.* et al. determined that autophagy induction in neutrophils using rapamycin is sufficient to induce NETs even in the absence of other priming factors; whereas *McInturff A.M.* et al. found that rapamycin reduces LPS-dependent NET formation [30,31,32]. Evidence also points to mTOR-dependent regulation of NET formation through post-transcriptional control of hypoxia-inducible factor 1 alpha expression [32]. Meanwhile, the use of wortmannin, a phosphatidylinositol 3-kinase inhibitor that interferes with autophagosome formation, leads neutrophils to apoptosis rather than NETosis in response to PMA and lipopolysaccharide [29].

Despite current advances, the signaling mechanisms that control NET formation remain mostly uncharacterized. Further studies are needed to understand the distinct molecular pathways regulating NETosis and their implications for neutrophil-mediated biological functions in health and disease.

## 3. Neutrophil Extracellular Traps in Renal Disease

### 3.1. Acute Kidney Injury

Acute kidney injury (AKI), a frequent cause of nephrology consultation and mortality, is characterized by a rapid decline in glomerular filtration rate associated with a decrease in renal blood flow, inflammation, or nephrotoxicity [33]. Pathological presentations of AKI often include damaged tubules, dysfunctional renal vasculature, excessive inflammation, and immune cell infiltration [34,35,36]. Although neutrophils are well-known elements of pro-inflammatory responses, the exact mechanisms through which neutrophils contribute to AKI are still debatable. However, late evidence involves NET release in the pathogenesis of AKI that results from ischemia-reperfusion injury and hemolytic uremic syndrome (HUS) [34,35,36,37,38,39].

Ischemic AKI boosts levels of circulating and localized NETs and histones; as well as PAD4 expression in the affected kidneys [11,29,30]. *Raup-Konsavage W.M.* et al. demonstrated that PAD4 expressing cells are mostly neutrophils that aggregate in peritubular capillaries, interstitial space, and renal tubules after ischemia-reperfusion injury [36,37]. NETs induce tubular epithelial cell death, promote clotting in peritubular capillaries via platelet-neutrophil interactions, and prime other neutrophils to undergo NETosis [11,38]. All these events sustain hypoxia and enhance tissue damage. Interestingly, PAD4 inhibition using pharmacological or genetic approaches protects from AKI in animal models due to a decrease in inflammation and NET formation. Meanwhile, degradation of NETs by DNase I or anti-histone IgG also reduces renal injury; underscoring the importance of NET formation in the pathogenesis of ischemic AKI [11,37]. Tubular necrosis and NET formation also augment remote organ dysfunction, a common feature of severe AKI, through the release of circulating histones and cytokines [11].

In HUS, a vascular disease caused by the Shiga toxin of enterohemorrhagic bacteria, neutrophils mediate an inflammatory response that is essential for the progression of hemolytic anemia, thrombocytopenia, and acute renal failure characteristic of the disease. NETs promote renal failure during HUS by contributing to the inflammatory response and thrombosis in the microvasculature. Furthermore, plasma from HUS patients contains increased levels of circulating cell free-DNA and nucleosomes compared to healthy subjects [39]. Neutrophils from HUS patients also exhibit a higher capacity to undergo spontaneous NETosis.

### 3.2. Lupus Nephritis

NETosis is a pathogenic feature of Systemic Lupus Erythematosus (SLE) frequently associated with active periods of the disease and production of autoantibodies [40]. Remarkably, defective NET removal in the sera of SLE patients correlates with signs of active nephritis such as proteinuria, decreased albumin levels, and lower creatinine clearance rates [41,42]. Neutrophils from patients with SLE spontaneously generate NETs that are more oxidized and immunogenic than those released by neutrophils from healthy subjects [43]. Moreover, high levels of NETs in the skin, kidney, and bone marrow reinforce their direct role in SLE-associated organ dysfunction [44]. The presence of NETs in renal biopsies from SLE patients with nephritis further supports their contribution to kidney damage [45].

DNase I mutations are present in some patients with SLE and could partially explain NET persistence in the sera of these patients [46]. Apoptotic or necrotic material from NETs is considered the primary source of self-antigens responsible for promoting an autoimmune state and exacerbating tissue injury [47]. Autoantibodies contribute to NET persistence by blocking their degradation and increasing complement activation, whereas the presence of NET components and autoantibodies directed against them tightly reflects SLE activity [48,49]. Immune complexes binding matrix metalloproteinase 2, a NET component, enhance NET release and matrix metalloproteinase 2 activity in serum from SLE patients [50]. Similarly, the LL37 peptide found in NETs associates with the immunogenic response to cell-free DNA in these patients [51]. Citrullinated histone H1 and anti-citrullinated H1 autoantibodies present in SLE patients also correlate with disease activity [52].

Additionally, SLE patients possess a distinct neutrophil subpopulation known as low-density granulocytes [53]. Low-density granulocytes are prone to pro-inflammatory cytokine release and spontaneous NET formation. These cells stimulate the production of interferon alpha (IFN-α), a central factor in the development of SLE pathogenesis, in plasmacytoid dendritic cells (pDCs) [51,54]. DNA and NET-associated proteins released by low-density granulocytes activate TLR9 in pDCs to induce IFN-α production [51]. Moreover, neutrophils on NETosis alone produce IFN-α in response to circulating chromatin, LL37, and high mobility group box 1 [55,56]. IFN-α is cytotoxic to endothelial cells as it can disrupt endothelial differentiation and angiogenesis [53]. Therefore, IFN-α-mediated effects on mature endothelial cells and their precursors are harmful to the renal vasculature during SLE flares. The concerted activity of neutrophils and pDCs amplify immune dysregulation, inflammation, and tissue damage via IFNα [54]. Besides their effects over pDCs, NET components also activate B cells, and in return, anti-DNA complexes can further boost the activity of pDCs and B cells, creating a detrimental autoimmune cycle [57].

Additionally, excessive NET production by SLE neutrophils is linked to hypoxia, upregulation of the stress-response protein DNA Damage Inducible Transcript 4 (DDIT4/REDD1), and autophagy [58]. NETs released by active SLE neutrophils are rich in tissue factor and interleukin 17A; molecules that promote thrombosis, inflammation, tissue injury, and fibrosis in target organs including the kidney. In patients with lupus nephritis *Frangou E.* et al. observed NETs comprised of tissue factor within glomeruli and the tubulointerstitial compartment in proximity to the Bowman’s capsule; suggesting a possible role for these NETs in capsule disruption and crescent formation [58].

### 3.3. Small Vessel and ANCA-Associated Vasculitis

ANCAs against MPO and proteinase 3 are diagnostic tools in small vessel vasculitis (SVV), a systemic autoimmune disease. Both autoantigens are NET components and participate in a harmful feedback circuit responsible for increased cell adhesion, complement activation, and NET production [59,60]. Blood samples of SVV patients have higher levels of MPO and DNA; whereas NETs are associated with strong neutrophilic infiltrates in biopsies from patients with active disease [61]. Additionally, patients with active SVV, present high serum levels of IFN-α, a cytokine related to pDCs activated by NET components [61]. Although studies to determine the longitudinal relationship of NETs with SVV are still on the works, it is evident that patients with active SVV have higher levels of circulating NETs and an increased propensity to polymorphonuclear cell death in comparison with patients in remission and healthy controls [62].

ANCA-associated vasculitis (AAV) belongs to a group of immune vasculitides defined by necrotizing inflammation of small vessels and circulating ANCAs [63]. Although neutrophils from patients with AAV are less likely to undergo apoptosis, they exhibit spontaneous NET formation [64]. As in SLE, there is evidence of NETs in kidney biopsies of AVV patients [61,65]. AAV comprise three diseases: granulomatosis with polyangiitis, microscopic polyangiitis (MPA), and eosinophilic granulomatosis with polyangiitis [66]. MPA is an ANCA-associated pathology that affects small vessels, particularly in renal glomeruli [67]. NET components, notably MPO, act as autoantibodies that increase circulating ANCAs and promote the subsequent development of MPA [68]. Conversely, immunoglobulins from MPA patients further induce NET release, and their ability to do so correlates with ANCAs affinity for MPO. Additionally, DNAse I activity is lower in sera from MPA patients, a feature that impairs adequate NET degradation [69]. NET release and persistence are also responsible for the renal damage present in up to 90% of the patients with MPA [64]. Although it has been largely assumed that ANCAs induce NET formation in neutrophils, *Kraaij T.* et al. recently demonstrated that uncontrolled NET formation in AAV patients is independent of ANCA levels in serum [65]. Their findings also indicate that NETosis is higher in patients positive for MPO-ANCAs rather than PR3-ANCAs which suggests that neutrophils and NETs might have different roles in granulomatosis with polyangiitis versus MPA pathogenesis. Moreover, excessive NETosis in patients with AAV associates with active clinical disease instead of severe infection; highlighting the role of NETs in autoimmunity [65].

## 4. Mechanisms of NET-Associated Tissue Injury

Though NET release is a legitimate innate defense system, NET components are not specific, and their actions can indirectly promote tissue injury and autoimmunity. Consequently, dysregulated production or clearance of NETs is detrimental for the kidneys. A recent study demonstrates that NETs are potentially toxic and their renal presence contributes to glomerular injury [70]. Neutrophil degrading enzymes can directly induce apoptosis in endothelial cells and degrade the basement membrane; whereas tubulointerstitial injury reduces glomerular blood flow, creating a NETotic environment [11,34,38]. Epithelial tubular cells release histones in response to hypoxia and kidney injury, which in turn activate neutrophils to release more NETs and creates a pro-inflammatory cycle that continues to inflict damage in endothelial cells [11,70]. Indirectly, NETs promote vascular damage by activating the alternative complement pathway and contribute to kidney fibrosis by inducing endothelial to mesenchymal (EndMT) transformation [70]. NET-associated mediators of tissue injury in kidney disease such as DNA, histones, NE, autoantibodies, and complement activation will be addressed in the following sections (Figure 2).

### 4.1. DNA and Histones

NET components are cytotoxic, enhance inflammation, and promote coagulation. Cell-free DNA from NETs is a strong activator of thrombotic disorders since it induces platelet aggregation, promotes coagulation, inhibits fibrinolysis, and interferes with cloth stability [71]. Mitochondrial CG-rich DNA is a more potent pro-inflammatory stimulus than nuclear DNA, though both are NET components [28,43,72].

Histones are vital elements of NETs with bactericidal action [73,74]. Extracellular histones activate immune responses through nod-like receptor protein 3, TLR2, and TLR4 [75,76]. Moreover, extracellular histones are cytotoxic, cause endothelial dysfunction, and contribute to organ failure in sepsis [77]. During AKI, histones released during tissue damage aggravate renal failure, and in vitro, extracellular histones kill renal endothelial and tubular epithelial cells [76]. Strikingly, extracellular histones increase ROS production and recruit neutrophils, creating a loop in favor of NET production [75]. Besides endothelial, renal, and neutrophilic action, histones induce thrombosis in platelets by promoting thrombin generation, platelet aggregation, prothrombinase activity, and expression of P-selectin, phosphatidylserine, and Factor V [78].

### 4.2. Neutrophil Elastase

A recent study by *Pieterse E.* et al. demonstrates a link between NE, vascular leakage, and profibrotic processes in NET-associated inflammatory conditions [70]. Excessive NETosis exceeds the phagocytic capacity of endothelial cells and results in VE-cadherin proteolysis, loss of intercellular junctions, development of vascular leakage, and EndMT induction through β-catenin signaling. EndMT has a prominent role in the development and progression of kidney injury through changes in cell shape, polarity, motility, and collagen production that contribute to interstitial fibrosis [79,80]. Glomerular presence of NETs and NET components such as NE and citrullinated histone H3 associates not only with EndMT but also with the degree of proteinuria [70,81,82,83]. The integrity of endothelial cell-cell contacts is mostly mediated through VE-cadherin and is a prerequisite for maintaining a restrictive endothelial barrier [84]. In conclusion, NETs play a role in the development of edema and proteinuria by increasing vascular permeability through NE-mediated loss of VE-cadherin.

### 4.3. Autoantibodies

Alterations in the underlying properties of autoantibodies before the onset of renal diseases are common, though kidneys can also become vulnerable to autoantibody deposition and binding. As described in previous sections, NET components such as histones, dsDNA, and neutrophil granular proteins act as ubiquitously present self-antigens in a multitude of renal diseases. Moreover, all described autoantigens for ANCAs are NET components [85]. These neutrophil-derived molecules need to be released into the extracellular environment for antibodies to recognize them as antigens. When autoantibody effector mechanisms fail to eliminate NET-associated antigens properly, inflammatory tissue damage and a new release of intracellular antigens occur. This dynamic process creates a destructive cycle wherein each disease flare elicits new autoantibody production and tissue injury [86]. B cell and myeloid cell activation via TLRs and immunoglobulin Fc receptors expressed on all hematopoietic cells further intensifies the pathogenic inflammatory response and contributes to its persistence [87,88,89,90]. NETs also increase T cell-mediated antigen responses and activate B cells to induce immunoglobulin and antibody production [44].

Although autoantibodies cause cellular damage and immune activation through diverse mechanisms, immune complex formation is a significant component of autoantibody-induced pathology. Immune complexes, molecules that result from an antibody binding to a soluble antigen, activate effector cells through Fc receptors and elicit phagocytosis, endocytosis of IgG-opsonized particles, the release of inflammatory mediators, and cytotoxicity [91]. Circulating levels of immune complexes containing DNA and IgG anti-dsDNA antibodies are nephritogenic [92,93]. The presence of charged residues within autoantibody heavy chain CDR regions, particularly CDR3, as well as the isotype and subclass of the antibody determine the risk of clinical nephritis [92,94,95,96]. Immobilized immune complexes also promote NET release through FcγRIIIB and macrophage antigen 1 activation, followed by downstream phosphorylation of p38 and extracellular signal-regulated protein kinases 1 and 2 [97]. As with other stimuli, this effect requires ROS production by NADPH oxidase and MPO activity.

Autoantibody-mediated tissue injury also relies upon the activation of the complement cascade via the classical pathway, which depends on target-bound antibodies, C1q, and activation of the C3-convertase C2bC4b [49,98]. The complement system leads to membrane attack complex formation in the plasma membrane of target cells through a series of serine-protease cascade reactions and consecutive cleavage of complement proteins [99]. The membrane attack complex promotes cell lysis by creating a pore that allows small molecules and metabolites to diffuse freely. Though complement activation and membrane attack complex formation aim at killing pathogens, they can also affect host cells and tissues if the complement system is uncontrolled.

### 4.4. The Complement System

Evidence supporting the relevance of the complement system for NETosis comes from neutrophils derived from C3- and C3a receptor-deficient mice, which are unable to form NETs [26,100]. Moreover, complement protein mediated opsonization initiated through classical or alternative pathways is vital for NET release; hence a pathogens ability to induce NETosis is inversely associated with its capacity to circumvent complement activation and opsonization [97,101,102]. In addition to C3, the anaphylatoxin C5a also enhances NET release by recruiting and then priming neutrophils through upregulation of TLRs and complement receptors [97,103,104,105].

NETosis, complement activation, and coagulation belong to a triangular, interconnected pathway [98]. TLR4 activation in platelets promotes cell adhesion to neutrophils; an association that heightens intravascular NET production and contributes to thromboinflammatory lesions and tissue damage [22]. Platelets induce NET release by presenting high mobility group box 1 proteins to receptors for advanced glycation end products in polymorphonuclear cells [25,106]. Besides their association with platelets, NETs also recruit red blood cells, promote thrombin deposition, and work as scaffolds during thrombus formation [71,107].

## 5. Therapeutic Interventions

As reviewed in previous sections, NETosis plays a significant role in the pathogenesis and progression of various inflammatory diseases that target the kidney [11,12,13]. Consequently, strategies aimed at NET-associated mechanisms of tissue injury could be helpful in ameliorating the severity of renal diseases. Since NET composition and their release mechanisms appear to be dependent on each disease’s unique pathogenesis, the development of a one-size-fits-all therapeutic intervention might be unsuitable to prevent NET formation while still maintaining neutrophil-mediated immune protection against infections [41,42,43,44]. For this reason, therapeutic interventions should be individually customized to each disease type.

The therapeutic potential of novel selective inhibitors of PAD4, the enzyme critical for protein citrullination and NET formation, is currently being explored in preclinical models of cancer-associated kidney injury and autoimmune disease; and it is expected that the efficacy of several such molecules will soon be tested in phase I/II clinical trials [34,37,108]. Although several PAD inhibitors have been characterized, most of these compounds are somewhat ineffective [109]. Irreversible haloacetamidine compounds including F- and Cl-amidine are active against PAD4 both, in vitro and in vivo [110]. Selective PAD4 inhibitors confirm the key role of PAD4 in histone citrullination, NET formation and ANCA production [111,112]. Moreover, administration of PAD4 inhibitors such as 2-chloramidine, YW3-56 and GSK484 before ischemia-reperfusion or in animal models of cancer-associated kidney injury significantly decreases renal damage, necrosis, congestion, and systemic inflammation [34,37,108]. Additionally, the use of the PAD inhibitors Cl-amidine and BB-Cl-amidine in lupus-prone mice modulates NET formation, reduces IFN-regulated gene expression and protects against lupus-dependent damage to the vasculature, kidneys and skin [113]. Despite current advances, the development of more potent, efficient and safe PAD4-specific inhibitors for clinical use remains crucial to translate experimental finding from animal models to human subjects [35].

Besides PAD4 inhibitors, the effects of the platelet inhibitor clopidrogel and the NE inhibitor sivelestat have also been studied in vitro and in vivo [38,83]. Platelet inhibition with clopidrogel reduces NET formation and kidney injury after ischemia-reperfusion in mice; whereas sivelestat decreases proteinuria, creatinine levels and glomerular damage in a nephritis rat model [38,83]. Inhibitors of nitric oxide production and oxidative stress also reduce NET release, suggesting they could be used to reduce NET-associated renal damage in vivo [114].

Dysregulated IFN production is influenced by NET release and subsequent activation of pDCs [51]. Clinical trials targeting NETosis and IFN-dependent pathways have been performed in SLE patients. Rontalizumab is a humanized IgG monoclonal antibody that neutralizes IFN-α. Patients with lupus were treated with rontalizumab in a phase one study that determined the antibody was safe and could reduce the expression of IFN-regulated genes; whereas similar results were obtained in a different trial evaluating the anti-IFNα monoclonal antibody sifalumimab [115,116]. Taken together, these results propose that modulation of IFN-dependent responses is a promising strategy to control autoimmune-mediated kidney damage. In neutrophils isolated from SLE patients, vitamin D reduces NET formation and endothelial cell apoptosis [117,118]. Moreover, low serum levels of 25(OH)D in patients with SLE associates with proteinuria and urinary vitamin D-binding protein [119]. Unfortunately, addressing vitamin D deficiency in small cohorts of SLE patients has no significant effect on immune markers or disease activity [120,121].

NET degradation, rather than NET formation, is involved in SLE pathogenesis and lupus nephritis [42]. In an experimental model of SLE, DNase I and dexamethasone treatment diminished proteinuria and serum creatinine, and improved renal histopathology [122]. Interestingly, sera from SLE patients contain DNase I inhibitors or present elevated titers of NET-binding antibodies; mechanisms that protect NETs from DNase I degradation [42,123,124]. Evidence suggests that lower DNAse I activity within pathological kidneys suppresses elimination of DNA debris; whereas DNAse I treatment protects kidneys from ischemia-induced AKI [125]. DNase I administration in animal models of ischemia-reperfusion ameliorates histologic characteristics of renal injury, improves renal function, reduces hypoxia and activates regeneration [125]. Therapeutic agents that target NET degradation, like DNase I, should be further explored to reduce NET-associated tissue damage in lupus nephritis and AKI. Additionally, intravenous human sulfo-immunoglobulin (IVIG-S) therapy, prepared from γ-globulins from healthy blood donors, reduces NET formation in vitro and in vivo, ANCA-MPO titers, and AAV development in a rat model [126]. Although IVIG therapy has been previously used to successfully treat autoimmune diseases, agammaglobulinemia, and severe infections in humans, prospective clinical trials are still needed to demonstrate the efficacy of IVIG therapy in AAV patients [126,127,128].

Characterization of neutrophil and NET-associated markers could also be useful to diagnose inflammatory pathologies affecting the kidney. Markers of NETosis previously described in human kidney diseases are shown in Table 1.

## 6. Conclusions

The field of NETosis has grown exponentially since 2004 when *Brinkmann V.* et al. described NETs as extracellular fibers comprised of chromatin and granule proteins released by neutrophils and able to bind bacteria [9]. Since then, NETosis has been extensively studied in vitro, in animal models, and human disease. Though the conclusions drawn by these studies are as diverse as the studies themselves, it is now evident that NETosis is more than a simple host-defense mechanism [9,13]. NETosis drives pathophysiological conditions associated with sterile inflammation and autoimmunity, and it is responsible for targeting the kidneys to promote the development of acute and chronic forms of renal injury [8,13,48,71]. NET-associated mechanisms play a role in AKI, HUS, lupus nephritis, SVV, and AAV; albeit the dynamics behind NET formation and clearance are not yet fully understood [34,35,36,37,38,39,40,59,60,63]. Neutrophils and NETosis could also be modulated in less explored pathological conditions such as chronic kidney disease and renal transplant. Hence, studies addressing the impact that dialysis, transplantation, immunosuppression, and high titers of donor-specific antibodies have on neutrophil biology and their ability to induce NET-mediated tissue damage are urgently needed.

NET formation is an important therapeutic target for the management of multiple human disorders, including renal diseases [113,115,116]. Understanding the molecular mechanisms behind NETosis and its relationship with apoptosis will also aid in the developing of tailored therapeutic strategies while reducing the risks of adverse side effects [24,29,64]. An important aspect that must also be considered is the crosstalk of neutrophils and NETs with other cell types such as platelets and endothelial cells, and how this interplay modulates biological outcomes in both, health and disease [70,71,78].

## Figures and Tables

**Figure 1 medicina-55-00431-f001:**
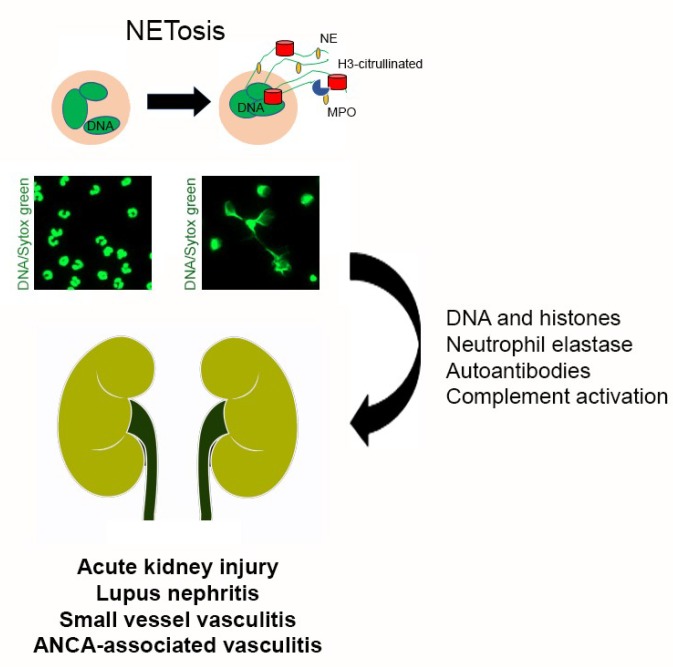
NETosis in Renal Disease. Neutrophil activation can lead to the release of NETs, mesh-like structures comprised of DNA and polymorphonuclear proteins such as myeloperoxidase (MPO), neutrophil elastase (NE) and citrullinated histones H3. NETs produce tissue damage through DNA, histones, neutrophil elastase, autoantibodies and complement activation. Moreover, NETs are involved in autoimmune and inflammatory diseases that damage the kidney such as acute kidney injury, lupus nephritis, small vessel and ANCA-associated vasculitis.

**Figure 2 medicina-55-00431-f002:**
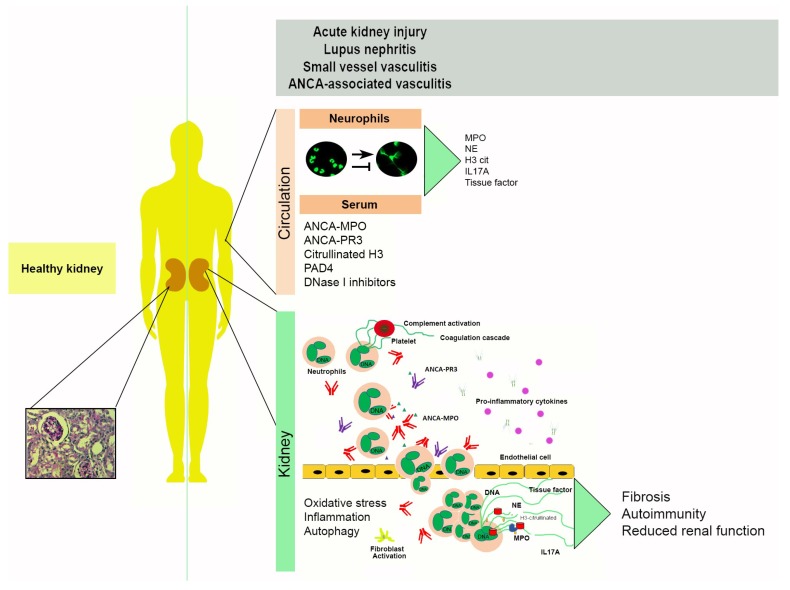
NET-associated mechanisms of tissue damage in kidney disease. The balance between NET release/NET degradation is relevant in the establishment and progression of multiple kidney diseases including acute kidney injury (AKI), lupus nephritis, small vessel vasculitis (SVV) and ANCA-associated vasculitis (AAV). *Circulation*: Netting neutrophils are in the circulation, along with decondensed DNA and NETosis markers detected in serum such as ANCA-MPO, ANCA-PR3, citrullinated histone H3, peptidyl arginine deiminase 4 (PAD4), and DNase I inhibitors that impair NET degradation. *Kidney*: Neutrophils can infiltrate the kidney in response to pro-inflammatory stimuli. Here, they can release NETs and NET components such as histones that are cytotoxic and promote pro-fibrotic phenotypes. Oxidative stress, inflammation, protein citrullination, and autophagy are molecular mechanisms closely linked to NET formation. Indirect tissue injury by NETs can result in fibrosis, autoimmunity, and a decrease in renal function.

**Table 1 medicina-55-00431-t001:** NETosis markers in human kidney disease.

Kidney Disease	NETosis Markers	References
Acute Kidney Injury	NE Citrullinated histone H3	[11]
	Circulating DNA Nucleosomes	[39]
Lupus Nephritis	Cell free DNA NET protecting antibodies	[41,42]
	NE MPO Histone H2A	[45]
Small vessels and ANCA-associated vasculitis	Nucleosomes MPO	[61,68]
	MPO Histone H2A Histone H2B PAD4 Citrullinated histone H3	[66]
	MPO Citrullinated histone H3	[69]

Abbreviations: *NE*, neutrophil elastase; *MPO*, myeloperoxidase; *PAD4*, peptidylarginasedeiminase 4; *ANCA*, anti-neutrophil cytoplasm antibodies.

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
