# Peer review of "Neutrophil Extracellular Traps in the Establishment and Progression of Renal Diseases"

_medicina, 2019, doi:10.3390/medicina55080431_

Round 1

Reviewer 1 Report

The manuscript is well written but lacks a few important features.

The authors should consider including some latest papers by Nakawaza group. https://doi.org/10.1016/ j.kint.2018.08.035

The section on therapeutic interventions should be included.

The authors should include a schematic with the kidney disease and NETs model.

Consider including a table with kidney disease, type of NETs, and intervention.

Author Response

Response to Reviewer 1

Dear reviewer, we appreciate your comments. The following changes have been made to the manuscript:

1- We included the paper that you suggested (https://doi.org/10.1016/ j.kint.2018.08.035), as well as other relevant papers that were initially left out of the review.

2-     A section on therapeutic interventions was included.

3-     Figure 2 was added to address kidney disease and NETs model.

4-  Table 1 was also added to report kidney disease and associated NETosis markers in humans.

Reviewer 2 Report

The manuscript of Salazar-Gonzalez et al is really easy to read and understand. Nevertheless, there is need for some improvements in order to be proper for publication.

The lines 51-52, 101-106 and section 4. Mechanisms of NET-associated tissue injury are missing references.

In 3.2. Lupus nephritis section you should include and comment REDD1/autophagy pathway promotes thromboinflammation and fibrosis in human systemic lupus erythematosus (SLE) through NETs decorated with tissue factor (TF) and interleukin-17A (IL-17A), Ann Rheum Dis. 2019 Feb;78(2):238-248. doi: 10.1136/annrheumdis-2018-213181 by Frangou et al.

In general, there are newer studies (references) in the field that should be added throughout the review.

The 5. Conclusions and future perspectives section must be extended. The authors have to discuss the existing literature more and give their opinion on the subject.

Author Response

Response to Reviewer 2

Dear reviewer, thank you for your time and your comments. The following changes have been made to the manuscript:

1-     Missing references (lines 51-52, 101-106 and section 4. Mechanisms of NET-associated tissue injury) were updated in the manuscript.

2-     In section 3.2. Lupus nephritis, we added a comment on REDD1/autophagy pathway promotes thromboinflammation and fibrosis in human systemic lupus erythematosus (SLE) through NETs decorated with tissue factor (TF) and interleukin-17A (IL-17A), Ann Rheum Dis. 2019 Feb;78(2):238-248. doi: 10.1136/annrheumdis-2018-213181 by Frangou et al. Additionally, we included more information regarding autophagy and NETosis that was missing.

3-  In response to your observation, newer studies in the field were added throughout the review.

4-     In response to comment 4, we added a new section (5. Therapeutic interventions) where we try to address what has been done to target NETosis in renal disease, as well as future perspectives. Section 6 was changed to conclusions only. In this section, we tried to summarize the purpose of the review, as well as give our opinion on the subject. We hope that these changes improved the quality of the manuscript.